# Mutual Mean-Teaching: Pseudo Label Refinery for Unsupervised Domain Adaptation on Person Re-identification

**Yixiao Ge, Dapeng Chen & Hongsheng Li**
The Chinese University of Hong Kong
{yxge@link,hsli@ee}.cuhk.edu.hk

## Abstract

Person re-identification (re-ID) aims at identifying the same persons' images across different cameras. However, domain diversities between different datasets pose an evident challenge for adapting the re-ID model trained on one dataset to another one. State-of-the-art unsupervised domain adaptation methods for person re-ID transferred the learned knowledge from the source domain by optimizing with pseudo labels created by clustering algorithms on the target domain. Although they achieved state-of-the-art performances, the inevitable label noise caused by the clustering procedure was ignored. Such noisy pseudo labels substantially hinders the model's capability on further improving feature representations on the target domain. In order to mitigate the effects of noisy pseudo labels, we propose to softly refine the pseudo labels in the target domain by proposing an unsupervised framework, Mutual Mean-Teaching (MMT), to learn better features from the target domain via off-line refined hard pseudo labels and on-line refined soft pseudo labels in an alternative training manner. In addition, the common practice is to adopt both the classification loss and the triplet loss jointly for achieving optimal performances in person re-ID models. However, conventional triplet loss cannot work with softly refined labels. To solve this problem, a novel soft softmax-triplet loss is proposed to support learning with soft pseudo triplet labels for achieving the optimal domain adaptation performance. The proposed MMT framework achieves considerable improvements of **14.4%**, **18.2%**, **13.4%** and **16.4%** mAP on Market-to-Duke, Duke-to-Market, Market-to-MSMT and Duke-to-MSMT unsupervised domain adaptation tasks. [1]

## 1 Introduction

Person re-identification (re-ID) aims at retrieving the same persons' images from images captured by different cameras. In recent years, person re-ID datasets with increasing numbers of images were proposed to facilitate the research along this direction. All the datasets require time-consuming annotations and are keys for re-ID performance improvements. However, even with such large-scale datasets, for person images from a new camera system, the person re-ID models trained on existing datasets generally show evident performance drops because of the domain gaps. Unsupervised Domain Adaptation (UDA) is therefore proposed to adapt the model trained on the source image domain (dataset) with identity labels to the target image domain (dataset) with no identity annotations.

State-of-the-art UDA methods (Song et al., 2018; Zhang et al., 2019b; Yang et al., 2019) for person re-ID group unannotated images with clustering algorithms and train the network with clustering-generated pseudo labels. Although the pseudo label generation and feature learning with pseudo labels are conducted alternatively to refine the pseudo labels to some extent, the training of the neural network is still substantially hindered by the inevitable label noise. The noise derives from the limited transferability of source-domain features, the unknown number of target-domain identities, and the imperfect results of the clustering algorithm. The refinery of noisy pseudo labels has crucial influences to the final performance, but is mostly ignored by the clustering-based UDA methods.

---

[1]Code is available at https://github.com/yxgeee/MMT.

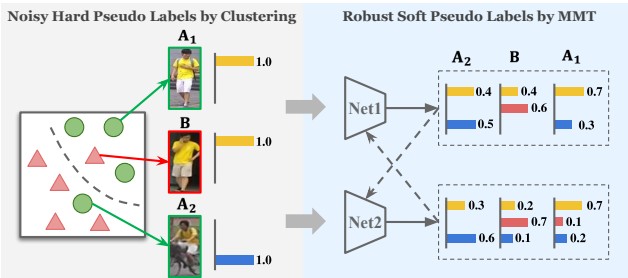

Figure 1: Person image $A_1$ and $A_2$ belong to the same identity while $B$ with similar appearance is from another person. However, clustering-generated pseudo labels in state-of-the-art Unsupervised Domain Adaptation (UDA) methods contain much noise that hinders feature learning. We propose pseudo label refinery with on-line refined soft pseudo labels to effectively mitigate the influence of noisy pseudo labels and improve UDA performance on person re-ID.

To effectively address the problem of noisy pseudo labels in clustering-based UDA methods (Song et al., 2018; Zhang et al., 2019b; Yang et al., 2019) (Figure 1), we propose an unsupervised Mutual Mean-Teaching (MMT) framework to effectively perform pseudo label refinery by optimizing the neural networks under the joint supervisions of off-line refined hard pseudo labels and on-line refined soft pseudo labels. Specifically, our proposed MMT framework provides robust soft pseudo labels in an on-line peer-teaching manner, which is inspired by the teacher-student approaches (Tarvainen & Valpola, 2017; Zhang et al., 2018b) to simultaneously train two same networks. The networks gradually capture target-domain data distributions and thus refine pseudo labels for better feature learning. To avoid training error amplification, the temporally average model of each network is proposed to produce reliable soft labels for supervising the other network in a collaborative training strategy. By training peer-networks with such on-line soft pseudo labels on the target domain, the learned feature representations can be iteratively improved to provide more accurate soft pseudo labels, which, in turn, further improves the discriminativeness of learned feature representations.

The classification and triplet losses are commonly adopted together to achieve state-of-the-art performances in both fully-supervised (Luo et al., 2019) and unsupervised (Zhang et al., 2019b; Yang et al., 2019) person re-ID models. However, the conventional triplet loss (Hermans et al., 2017) cannot work with such refined soft labels. To enable using the triplet loss with soft pseudo labels in our MMT framework, we propose a novel soft softmax-triplet loss so that the network can benefit from softly refined triplet labels. The introduction of such soft softmax-triplet loss is also the key to the superior performance of our proposed framework. Note that the collaborative training strategy on the two networks is only adopted in the training process. Only one network is kept in the inference stage without requiring any additional computational or memory cost.

The contributions of this paper could be summarized as three-fold. (1) We propose to tackle the label noise problem in state-of-the-art clustering-based UDA methods for person re-ID, which is mostly ignored by existing methods but is shown to be crucial for achieving superior final performance. The proposed Mutual Mean-Teaching (MMT) framework is designed to provide more reliable soft labels. (2) Conventional triplet loss can only work with hard labels. To enable training with soft triplet labels for mitigating the pseudo label noise, we propose the soft softmax-triplet loss to learn more discriminative person features. (3) The MMT framework shows exceptionally strong performances on all UDA tasks of person re-ID. Compared with state-of-the-art methods, it leads to significant improvements of **14.4%**, **18.2%**, **13.4%**, **16.4%** mAP on Market-to-Duke, Duke-to-Market, Market-to-MSMT, Duke-to-MSMT re-ID tasks.

## 2 RELATED WORK

**Unsupervised domain adaptation (UDA) for person re-ID.** UDA methods have attracted much attention because their capability of saving the cost of manual annotations. There are three main categories of methods. The first category of clustering-based methods maintains state-of-the-art performance to date. (Fan et al., 2018) proposed to alternatively assign labels for unlabeled training samples and optimize the network with the generated targets. (Lin et al., 2019) proposed a bottom-

up clustering framework with a repelled loss. (Yang et al., 2019) introduced to assign hard pseudo labels for both global and local features. However, the training of the neural network was substantially hindered by the noise of the hard pseudo labels generated by clustering algorithms, which was mostly ignored by existing methods. The second category of methods learns domain-invariant features from style-transferred source-domain images. SPGAN (Deng et al., 2018) and PTGAN (Wei et al., 2018) transformed source-domain images to match the image styles of the target domain while maintaining the original person identities. The style-transferred images and their identity labels were then used to fine-tune the model. HHL (Zhong et al., 2018) learned camera-invariant features with camera style transferred images. However, the retrieval performances of these methods deeply relied on the image generation quality, and they did not explore the complex relations between different samples in the target domain. The third category of methods attempts on optimizing the neural networks with soft labels for target-domain samples by computing the similarities with reference images or features. ENC (Zhong et al., 2019) assigned soft labels by saving averaged features with an exemplar memory module. MAR (Yu et al., 2019) conducted multiple soft-label learning by comparing with a set of reference persons. However, the reference images and features might not be representative enough to generate accurate labels for achieving advanced performances.

**Generic domain adaptation methods for close-set recognition.** Generic domain adaptation methods learn features that can minimize the differences between data distributions of source and target domains. Adversarial learning based methods (Zhang et al., 2018a; Tzeng et al., 2017; Ghifary et al., 2016; Bousmalis et al., 2016; Tzeng et al., 2015) adopted a domain classifier to dispel the discriminative domain information from the learned features in order to reduce the domain gap. There also exist methods (Tzeng et al., 2014; Long et al., 2015; Yan et al., 2017; Saito et al., 2018; Ghifary et al., 2016) that minimize the Maximum Mean Discrepancy (MMD) loss between source- and target-domain distributions. However, these methods assume that the classes on different domains are shared, which is not suitable for unsupervised domain adaptation on person re-ID.

**Teacher-student models** have been widely studied in semi-supervised learning methods and knowledge/model distillation methods. The key idea of teacher-student models is to create consistent training supervisions for labeled/unlabeled data via different models' predictions. Temporal ensembling (Laine & Aila, 2016) maintained an exponential moving average prediction for each sample as the supervisions of the unlabeled samples, while the mean-teacher model (Tarvainen & Valpola, 2017) averaged model weights at different training iterations to create the supervisions for unlabeled samples. Deep mutual learning (Zhang et al., 2018b) adopted a pool of student models instead of the teacher models by training them with supervisions from each other. However, existing methods with teacher-student mechanisms are mostly designed for close-set recognition problems, where both labeled and unlabeled data share the same set of class labels and could not be directly utilized on unsupervised domain adaptation tasks of person re-ID.

**Generic methods for handling noisy labels** can be classified into four categories. Loss correction methods (Patrini et al., 2017; Vahdat, 2017; Xiao et al., 2015) tried to model the noise transition matrix, however, such matrix is hard to estimate in real-world tasks, *e.g.* unsupervised person re-ID with noisy pseudo labels obtained via clustering algorithm. (Veit et al., 2017; Lee et al., 2018; Li et al., 2017; Han et al., 2019) attempted to correct the noisy labels directly, while the clean set required by such methods limits their generalization on real-world applications. Noise-robust methods designed robust loss functions against label noises, for instance, Mean Absolute Error (MAE) loss (Ghosh et al., 2017), Generalized Cross Entropy (GCE) loss (Zhang & Sabuncu, 2018) and Label Smoothing Regularization (LSR) (Szegedy et al., 2016). However, these methods did not study how to handle the triplet loss with noisy labels, which is crucial for learning discriminative feature representations on person re-ID. The last kind of methods which focused on refining the training strategies is mostly related to our method. Co-teaching (Han et al., 2018) trained two collaborative networks and conducted noisy label detection by selecting on-line clean data for each other, Co-mining (Wang et al., 2019) further extended this method on the face recognition task with a re-weighting function for Arc-Softmax loss (Deng et al., 2019). However, the above methods are not designed for the open-set person re-ID task and could not achieve state-of-the-art performances under the more challenge unsupervised settings.

# 3 PROPOSED APPROACH

We propose a novel Mutual Mean-Teaching (MMT) framework for tackling the problem of noisy pseudo labels in clustering-based Unsupervised Domain Adaptation (UDA) methods. The label noise has important impacts to the domain adaptation performance but was mostly ignored by those methods. Our key idea is to conduct pseudo label refinery in the target domain by optimizing the neural networks with off-line refined hard pseudo labels and on-line refined soft pseudo labels in a collaborative training manner. In addition, the conventional triplet loss cannot properly work with soft labels. A novel soft softmax-triplet loss is therefore introduced to better utilize the softly refined pseudo labels. Both the soft classification loss and the soft softmax-triplet loss work jointly to achieve optimal domain adaptation performances.

Formally, we denote the source domain data as $\mathbb{D}_s = \{(\boldsymbol{x}_i^s, \boldsymbol{y}_i^s)|_{i=1}^{N_s}\}$, where $\boldsymbol{x}_i^s$ and $\boldsymbol{y}_i^s$ denote the $i$-th training sample and its associated person identity label, $N_s$ is the number of images, and $M_s$ denotes the number of person identities (classes) in the source domain. The $N_t$ target-domain images are denoted as $\mathbb{D}_t = \{\boldsymbol{x}_i^t|_{i=1}^{N_t}\}$, which are not associated with any ground-truth identity label.

## 3.1 CLUSTERING-BASED UDA METHODS REVISIT

State-of-the-art UDA methods (Fan et al., 2018; Lin et al., 2019; Zhang et al., 2019b; Yang et al., 2019) follow a similar general pipeline. They generally pre-train a deep neural network $F(\cdot|\boldsymbol{\theta})$ on the source domain, where $\boldsymbol{\theta}$ denotes current network parameters, and the network is then transferred to learn from the images in the target domain. The source-domain images' and target-domain images' features encoded by the network are denoted as $\{F(\boldsymbol{x}_i^s|\boldsymbol{\theta})\}|_{i=1}^{N_s}$ and $\{F(\boldsymbol{x}_i^t|\boldsymbol{\theta})\}|_{i=1}^{N_t}$ respectively. As illustrated in Figure 2 (a), two operations are alternated to gradually fine-tune the pre-trained network on the target domain. (1) The target-domain samples are grouped into pre-defined $M_t$ classes by clustering the features $\{F(\boldsymbol{x}_i^t|\boldsymbol{\theta})\}|_{i=1}^{N_t}$ output by the current network. Let $\tilde{\boldsymbol{y}}_i^t$ denotes the pseudo label generated for image $\boldsymbol{x}_i^t$. (2) The network parameters $\boldsymbol{\theta}$ and a learnable target-domain classifier $C^t : \boldsymbol{f}^t \rightarrow \{1, \cdots, M_t\}$ are then optimized with respect to an identity classification (cross-entropy) loss $\mathcal{L}_{id}^t(\boldsymbol{\theta})$ and a triplet loss (Hermans et al., 2017) $\mathcal{L}_{tri}^t(\boldsymbol{\theta})$ in the form of,

$$\mathcal{L}_{id}^t(\boldsymbol{\theta}) = \frac{1}{N_t} \sum_{i=1}^{N_t} \mathcal{L}_{ce}\left(C^t(F(\boldsymbol{x}_i^t|\boldsymbol{\theta})), \tilde{\boldsymbol{y}}_i^t\right), \tag{1}$$

$$\mathcal{L}_{tri}^t(\boldsymbol{\theta}) = \frac{1}{N_t} \sum_{i=1}^{N_t} \max\left(0, ||F(\boldsymbol{x}_i^t|\boldsymbol{\theta}) - F(\boldsymbol{x}_{i,p}^t|\boldsymbol{\theta})|| + m - ||F(\boldsymbol{x}_i^t|\boldsymbol{\theta}) - F(\boldsymbol{x}_{i,n}^t|\boldsymbol{\theta})||\right), \tag{2}$$

where $|| \cdot ||$ denotes the $L^2$-norm distance, subscripts $_{i,p}$ and $_{i,n}$ indicate the hardest positive and hardest negative feature index in each mini-batch for the sample $\boldsymbol{x}_i^t$, and $m = 0.5$ denotes the triplet distance margin. Such two operations, pseudo label generation by clustering and feature learning with pseudo labels, are alternated until the training converges. However, the pseudo labels generated in step (1) inevitably contain errors due to the imperfection of features as well as the errors of the clustering algorithms, which hinder the feature learning in step (2). To mitigate the pseudo label noise, we propose the Mutual Mean-Teaching (MMT) framework together with a novel soft softmax-triplet loss to conduct the pseudo label refinery.

## 3.2 MUTUAL MEAN-TEACHING (MMT) FRAMEWORK

### 3.2.1 SUPERVISED PRE-TRAINING FOR SOURCE DOMAIN

UDA task on person re-ID aims at transferring the knowledge from a pre-trained model on the source domain to the target domain. A deep neural network is first pre-trained on the source domain. Given the training data $\mathbb{D}_s$, the network is trained to model a feature transformation function $F(\cdot|\boldsymbol{\theta})$ that transforms each input sample $\boldsymbol{x}_i^s$ into a feature representation $F(\boldsymbol{x}_i^s|\boldsymbol{\theta})$. Given the encoded features, the identification classifier $C^s$ outputs an $M_s$-dimensional probability vector to predict the identities in the source-domain training set. The neural network is trained with a classification loss $\mathcal{L}_{id}^s(\boldsymbol{\theta})$ and a triplet loss $\mathcal{L}_{tri}^s(\boldsymbol{\theta})$ to separate features belonging to different identities. The overall loss is therefore calculated as

$$\mathcal{L}^s(\boldsymbol{\theta}) = \mathcal{L}_{id}^s(\boldsymbol{\theta}) + \lambda^s \mathcal{L}_{tri}^s(\boldsymbol{\theta}), \tag{3}$$

where $\mathcal{L}_{id}^s(\boldsymbol{\theta})$ and $\mathcal{L}_{tri}^s(\boldsymbol{\theta})$ are defined similarly to equation 1 and equation 2 but with ground-truth identity labels $\{\boldsymbol{y}_i^s|_{i=1}^{N_s}\}$, and $\lambda^s$ is the parameter weighting the two losses.

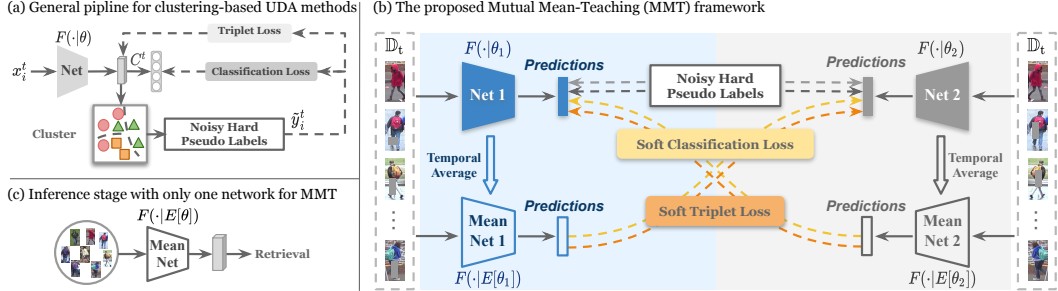

Figure 2: (a) The pipeline for existing clustering-based UDA methods on person re-ID with noisy hard pseudo labels. (b) Overall framework of the proposed Mutual Mean-Teaching (MMT) with two collaborative networks jointly optimized under the supervisions of off-line refined hard pseudo labels and on-line refined soft pseudo labels. A soft identity classification loss and a novel soft softmax-triplet loss are adopted. (c) One of the average models with better validated performance is adopted for inference as average models perform better than models with current parameters.

### 3.2.2 PSEUDO LABEL REFINERY WITH ON-LINE REFINED SOFT PSEUDO LABELS

Our proposed MMT framework is based on the clustering-based UDA methods with off-line refined hard pseudo labels as introduced in Section 3.1, where the pseudo label generation and refinement are conducted alternatively. However, the pseudo labels generated in this way are hard (*i.e.*, they are always of $100\%$ confidences) but noisy. In order to mitigate the pseudo label noise, apart from the off-line refined hard pseudo labels, our framework further incorporates on-line refined soft pseudo labels (*i.e.*, pseudo labels with $< 100\%$ confidences) into the training process.

Our MMT framework generates soft pseudo labels by collaboratively training two same networks with different initializations. The overall framework is illustrated in Figure 2 (b). The pseudo classes are still generated the same as those by existing clustering-based UDA methods, where each cluster represents one class. In addition to the hard and noisy pseudo labels, our two collaborative networks also generate on-line soft pseudo labels by network predictions for training each other. The intuition is that, after the networks are trained even with hard pseudo labels, they can roughly capture the training data distribution and their class predictions can therefore serve as soft class labels for training. However, such soft labels are generally not perfect because of the training errors and noisy hard pseudo labels in the first place. To avoid two networks collaboratively bias each other, the past temporally average model of each network instead of the current model is used to generate the soft pseudo labels for the other network. Both off-line hard pseudo labels and on-line soft pseudo labels are utilized jointly to train the two collaborative networks. After training, only one of the past average models with better validated performance is adopted for inference (see Figure 2 (c)).

We denote the two collaborative networks as feature transformation functions $F(\cdot|\boldsymbol{\theta}_1)$ and $F(\cdot|\boldsymbol{\theta}_2)$, and denote their corresponding pseudo label classifiers as $C_1^t$ and $C_2^t$, respectively. To simultaneously train the coupled networks, we feed the same image batch to the two networks but with separately random erasing, cropping and flipping. Each target-domain image can be denoted by $\boldsymbol{x}_i^t$ and $\boldsymbol{x'}_i^t$ for the two networks, and their pseudo label confidences can be predicted as $C_1^t(F(\boldsymbol{x}_i^t|\boldsymbol{\theta}_1))$ and $C_2^t(F(\boldsymbol{x'}_i^t|\boldsymbol{\theta}_2))$. One naïve way to train the collaborative networks is to directly utilize the above pseudo label confidence vectors as the soft pseudo labels for training the other network. However, in such a way, the two networks' predictions might converge to equal each other and the two networks lose their output independences. The classification errors as well as pseudo label errors might be amplified during training. In order to avoid error amplification, we propose to use the temporally average model of each network to generate reliable soft pseudo labels for supervising the other network. Specifically, the parameters of the temporally average models of the two networks at current iteration $T$ are denoted as $E^{(T)}[\boldsymbol{\theta}_1]$ and $E^{(T)}[\boldsymbol{\theta}_2]$ respectively, which can be calculated as

$$E^{(T)}[\boldsymbol{\theta}_1] = \alpha E^{(T-1)}[\boldsymbol{\theta}_1] + (1-\alpha)\boldsymbol{\theta}_1,$$
$$E^{(T)}[\boldsymbol{\theta}_2] = \alpha E^{(T-1)}[\boldsymbol{\theta}_2] + (1-\alpha)\boldsymbol{\theta}_2, \tag{4}$$

where $E^{(T-1)}[\boldsymbol{\theta}_1]$, $E^{(T-1)}[\boldsymbol{\theta}_2]$ indicate the temporal average parameters of the two networks in the previous iteration $(T-1)$, the initial temporal average parameters are $E^{(0)}[\boldsymbol{\theta}_1] = \boldsymbol{\theta}_1$, $E^{(0)}[\boldsymbol{\theta}_2] = \boldsymbol{\theta}_2$,

and $\alpha$ is the ensembling momentum to be within the range $[0, 1)$. The robust soft pseudo label supervisions are then generated by the two temporal average models as $C_1^t(F(\boldsymbol{x}_i^t|E^{(T)}[\boldsymbol{\theta}_1]))$ and $C_2^t(F(\boldsymbol{x}'_i^t|E^{(T)}[\boldsymbol{\theta}_2]))$ respectively. The soft classification loss for optimizing $\boldsymbol{\theta}_1$ and $\boldsymbol{\theta}_2$ with the soft pseudo labels generated from the other network can therefore be formulated as

$$\mathcal{L}_{sid}^t(\boldsymbol{\theta}_1|\boldsymbol{\theta}_2) = -\frac{1}{N_t} \sum_{i=1}^{N_t} \left( C_2^t(F(\boldsymbol{x}'_i^t|E^{(T)}[\boldsymbol{\theta}_2])) \cdot \log C_1^t(F(\boldsymbol{x}_i^t|\boldsymbol{\theta}_1)) \right),$$

$$\mathcal{L}_{sid}^t(\boldsymbol{\theta}_2|\boldsymbol{\theta}_1) = -\frac{1}{N_t} \sum_{i=1}^{N_t} \left( C_1^t(F(\boldsymbol{x}_i^t|E^{(T)}[\boldsymbol{\theta}_1])) \cdot \log C_2^t(F(\boldsymbol{x}'_i^t|\boldsymbol{\theta}_2)) \right). \tag{5}$$

The two networks' pseudo-label predictions are better dis-related by using other network's past average model to generate supervisions and can therefore better avoid error amplification.

Generalizing classification cross-entropy loss to work with soft pseudo labels has been well studied (Hinton et al., 2015), (Müller et al., 2019). However, optimizing triplet loss with soft pseudo labels poses a great challenge as no previous method has investigated soft labels for triplet loss. For tackling the difficulty, we propose to use softmax-triplet loss, whose hard version is formulated as

$$\mathcal{L}_{tri}^t(\boldsymbol{\theta}_1) = \frac{1}{N_t} \sum_{i=1}^{N_t} \mathcal{L}_{bce}\left( \mathcal{T}_i(\boldsymbol{\theta}_1), \mathbf{1} \right), \tag{6}$$

where

$$\mathcal{T}_i(\boldsymbol{\theta}_1) = \frac{\exp(\|F(\boldsymbol{x}_i^t|\boldsymbol{\theta}_1) - F(\boldsymbol{x}_{i,n}^t|\boldsymbol{\theta}_1)\|)}{\exp(\|F(\boldsymbol{x}_i^t|\boldsymbol{\theta}_1) - F(\boldsymbol{x}_{i,p}^t|\boldsymbol{\theta}_1)\|) + \exp(\|F(\boldsymbol{x}_i^t|\boldsymbol{\theta}_1) - F(\boldsymbol{x}_{i,n}^t|\boldsymbol{\theta}_1)\|)}. \tag{7}$$

Here $\mathcal{L}_{bce}(\cdot, \cdot)$ denotes the binary cross-entropy loss, $F(\boldsymbol{x}_i^t|\boldsymbol{\theta}_1)$ is the encoded feature for target-domain sample $\boldsymbol{x}_i^t$ by network 1, the subscripts $_{i,p}$ and $_{i,n}$ denote sample $\boldsymbol{x}_i^t$'s hardest positive and negative samples in the mini-batch, $\|F(\boldsymbol{x}_i^t|\boldsymbol{\theta}_1) - F(\boldsymbol{x}_{i,p}^t|\boldsymbol{\theta}_1)\|$ is the $L^2$-norm distance between sample $\boldsymbol{x}_i^t$ and its positive sample $\boldsymbol{x}_{i,p}^t$ to measure their similarity, and "$\mathbf{1}$" denotes the ground-truth that the positive sample $\boldsymbol{x}_{i,p}^t$ should be closer to the sample $\boldsymbol{x}_i^t$ than its negative sample $\boldsymbol{x}_{i,n}^t$. Given the two collaborative networks, we can utilize the one network's past temporal average model to generate soft triplet labels for the other network with the proposed soft softmax-triplet loss,

$$\mathcal{L}_{stri}^t(\boldsymbol{\theta}_1|\boldsymbol{\theta}_2) = \frac{1}{N_t} \sum_{i=1}^{N_t} \mathcal{L}_{bce}\left( \mathcal{T}_i(\boldsymbol{\theta}_1), \mathcal{T}_i\left( E^{(T)}[\boldsymbol{\theta}_2]\right) \right),$$

$$\mathcal{L}_{stri}^t(\boldsymbol{\theta}_2|\boldsymbol{\theta}_1) = \frac{1}{N_t} \sum_{i=1}^{N_t} \mathcal{L}_{bce}\left( \mathcal{T}_i(\boldsymbol{\theta}_2), \mathcal{T}_i\left( E^{(T)}[\boldsymbol{\theta}_1]\right) \right), \tag{8}$$

where $\mathcal{T}_i(E^{(T)}[\boldsymbol{\theta}_1])$ and $\mathcal{T}_i(E^{(T)}[\boldsymbol{\theta}_2])$ are the soft triplet labels generated by the two networks' past temporally average models. Such soft triplet labels are fixed as training supervisions. By adopting the soft softmax-triplet loss, our MMT framework overcomes the limitation of hard supervisions by the conventional triple loss (equation 2). It can be successfully trained with soft triplet labels, which are shown to be important for improving the domain adaptation performance in our experiments. Note that such a softmax-triplet loss was also studied in (Zhang et al., 2019a). However, it has never been used to generate soft labels and was not designed to work with soft pseudo labels before.

### 3.2.3 OVERALL LOSS AND ALGORITHM

Our proposed MMT framework is trained with both off-line refined hard pseudo labels and on-line refined soft pseudo labels. The overall loss function $\mathcal{L}(\boldsymbol{\theta}_1, \boldsymbol{\theta}_2)$ simultaneously optimizes the coupled networks, which combines equation 1, equation 5, equation 6, equation 8 and is formulated as,

$$\begin{aligned} \mathcal{L}(\boldsymbol{\theta}_1, \boldsymbol{\theta}_2) = \ & (1 - \lambda_{id}^t)(\mathcal{L}_{id}^t(\boldsymbol{\theta}_1) + \mathcal{L}_{id}^t(\boldsymbol{\theta}_2)) + \lambda_{id}^t(\mathcal{L}_{sid}^t(\boldsymbol{\theta}_1|\boldsymbol{\theta}_2) + \mathcal{L}_{sid}^t(\boldsymbol{\theta}_2|\boldsymbol{\theta}_1)) \\ & + (1 - \lambda_{tri}^t)(\mathcal{L}_{tri}^t(\boldsymbol{\theta}_1) + \mathcal{L}_{tri}^t(\boldsymbol{\theta}_2)) + \lambda_{tri}^t(\mathcal{L}_{stri}^t(\boldsymbol{\theta}_1|\boldsymbol{\theta}_2) + \mathcal{L}_{stri}^t(\boldsymbol{\theta}_2|\boldsymbol{\theta}_1)), \end{aligned} \tag{9}$$

where $\lambda_{id}^t, \lambda_{tri}^t$ are the weighting parameters. The detailed optimization procedures are summarized in Algorithm 1. The hard pseudo labels are off-line refined after training with existing hard pseudo labels for one epoch. During the training process, the two networks are trained by combining the off-line refined hard pseudo labels and on-line refined soft labels predicted by their peers with proposed soft losses. The noise and randomness caused by hard clustering, which lead to unstable training and limited final performance, can be alleviated by the proposed MMT framework.

**Require:** Target-domain data $\mathbb{D}_t$;
**Require:** Ensembling momentum $\alpha$ for equation 4, weighting factors $\lambda_{id}^t$, $\lambda_{tri}^t$ for equation 9;
**Require:** Initialize pre-trained weights $\boldsymbol{\theta}_1$ and $\boldsymbol{\theta}_2$ by optimizing with equation 3 on $\mathbb{D}_s$.
    **for** n in $[1, num\_epochs]$ **do**
        Generate hard pseudo labels $\tilde{\boldsymbol{y}}_i^t$ for each sample $\boldsymbol{x}_i^t$ in $\mathbb{D}_t$ by clustering algorithms.
        **for** each mini-batch $B \subset \mathbb{D}_t$, iteration $T$ **do**
          **1:** Generate soft pseudo labels from the collaborative networks by predicting $\mathcal{T}_{i \in B}(E^{(T)}[\boldsymbol{\theta}_1])$, $\mathcal{T}_{i \in B}(E^{(T)}[\boldsymbol{\theta}_2])$,
$C_1^t(F(\boldsymbol{x}_{i \in B}^t | E^{(T)}[\boldsymbol{\theta}_1]))$, $C_2^t(F(\boldsymbol{x'}_{i \in B}^t | E^{(T)}[\boldsymbol{\theta}_2]))$;
          **2:** Joint update parameters $\boldsymbol{\theta}_1$ & $\boldsymbol{\theta}_2$ by the gradient descent of the objective function equation 9;
          **3:** Update temporally average model weights $E^{(T+1)}[\boldsymbol{\theta}_1]$ & $E^{(T+1)}[\boldsymbol{\theta}_2]$ following equation 4.
        **end for**
    **end for**

**Algorithm 1:** Unsupervised Mutual Mean-Teaching (MMT) Training Strategy

# 4 EXPERIMENTS

## 4.1 DATASETS

We evaluate our proposed MMT on three widely-used person re-ID datasets, *i.e.*, Market-1501 (Zheng et al., 2015), DukeMTMC-reID (Ristani et al., 2016), and MSMT17 (Wei et al., 2018). The Market-1501 (Zheng et al., 2015) dataset consists of 32,668 annotated images of 1,501 identities shot from 6 cameras in total, for which 12,936 images of 751 identities are used for training and 19,732 images of 750 identities are in the test set. DukeMTMC-reID (Ristani et al., 2016) contains 16,522 person images of 702 identities for training, and the remaining images out of another 702 identities for testing, where all images are collected from 8 cameras. MSMT17 (Wei et al., 2018) is the most challenging and large-scale dataset consisting of 126,441 bounding boxes of 4,101 identities taken by 15 cameras, for which 32,621 images of 1,041 identities are spitted for training. For evaluating the domain adaptation performance of different methods, four domain adaptation tasks are set up, *i.e.*, Duke-to-Market, Market-to-Duke, Duke-to-MSMT and Market-to-MSMT, where only identity labels on the source domain are provided. Mean average precision (mAP) and CMC top-1, top-5, top-10 accuracies are adopted to evaluate the methods' performances.

## 4.2 IMPLEMENTATION DETAILS

### 4.2.1 TRAINING DATA ORGANIZATION

For both source-domain pre-training and target-domain fine-tuning, each training mini-batch contains 64 person images of 16 actual or pseudo identities (4 for each identity). Note that the generated hard pseudo labels for the target-domain fine-tuning are updated after each epoch, so the mini-batch of target-domain images needs to be re-organized with updated hard pseudo labels after each epoch. All images are resized to $256 \times 128$ before being fed into the networks.

### 4.2.2 OPTIMIZATION DETAILS

All the hyper-parameters of the proposed MMT framework are chosen based on a validation set of the Duke-to-Market task with $M_t = 500$ pseudo identities and IBN-ResNet-50 backbone. The same hyper-parameters are then directly applied to the other three domain adaptation tasks. We propose a two-stage training scheme, where ADAM optimizer is adopted to optimize the networks with a weight decay of 0.0005. Randomly erasing (Zhong et al., 2017b) is only adopted in target-domain fine-tuning.

**Stage 1: Source-domain pre-training.** We adopt ResNet-50 (He et al., 2016) or IBN-ResNet-50 (Pan et al., 2018) as the backbone networks, where IBN-ResNet-50 achieves better performances by integrating both IN and BN modules. Two same networks are initialized with ImageNet (Deng et al., 2009) pre-trained weights. Given the mini-batch of images, network parameters $\boldsymbol{\theta}_1$, $\boldsymbol{\theta}_2$ are updated independently by optimizing equation 3 with $\lambda^s = 1$. The initial learning rate is set to 0.00035 and is decreased to 1/10 of its previous value on the 40th and 70th epoch in the total 80 epochs.

**Stage 2: End-to-end training with MMT.** Based on pre-trained weights $\boldsymbol{\theta}_1$ and $\boldsymbol{\theta}_2$, the two networks are collaboratively updated by optimizing equation 9 with the loss weights $\lambda_{id}^t = 0.5$, $\lambda_{tri}^t = 0.8$. The temporal ensemble momentum $\alpha$ in equation 4 is set to 0.999. The learning rate is fixed to 0.00035 for overall 40 training epochs. We utilize *k-means* clustering algorithm and the number $M_t$ of pseudo classes is set as 500, 700, 900 for Market-1501 and DukeMTMC-reID, and 500, 1000, 1500, 2000 for MSMT17. Note that actual identity numbers in the target-domain training

| Methods | Market-to-Duke | | | | Duke-to-Market | | | |
|---|---|---|---|---|---|---|---|---|
| | mAP | top-1 | top-5 | top-10 | mAP | top-1 | top-5 | top-10 |
| PUL (Fan et al., 2018) (TOMM'18) | 16.4 | 30.0 | 43.4 | 48.5 | 20.5 | 45.5 | 60.7 | 66.7 |
| TJ-AIDL (Wang et al., 2018) (CVPR'18) | 23.0 | 44.3 | 59.6 | 65.0 | 26.5 | 58.2 | 74.8 | 81.1 |
| SPGAN (Deng et al., 2018) (CVPR'18) | 22.3 | 41.1 | 56.6 | 63.0 | 22.8 | 51.5 | 70.1 | 76.8 |
| HHL (Zhong et al., 2018) (ECCV'18) | 27.2 | 46.9 | 61.0 | 66.7 | 31.4 | 62.2 | 78.8 | 84.0 |
| CFSM (Chang et al., 2019) (AAAI'19) | 27.3 | 49.8 | - | - | 28.3 | 61.2 | - | - |
| BUC (Lin et al., 2019) (AAAI'19) | 27.5 | 47.4 | 62.6 | 68.4 | 38.3 | 66.2 | 79.6 | 84.5 |
| ARN (Li et al., 2018) (CVPR'18-WS) | 33.4 | 60.2 | 73.9 | 79.5 | 39.4 | 70.3 | 80.4 | 86.3 |
| UDAP (Song et al., 2018) (Arxiv'18) | 49.0 | 68.4 | 80.1 | 83.5 | 53.7 | 75.8 | 89.5 | 93.2 |
| ENC (Zhong et al., 2019) (CVPR'19) | 40.4 | 63.3 | 75.8 | 80.4 | 43.0 | 75.1 | 87.6 | 91.6 |
| UCDA-CCE (Qi et al., 2019) (ICCV'19) | 31.0 | 47.7 | - | - | 30.9 | 60.4 | - | - |
| PDA-Net (Li et al., 2019) (ICCV'19) | 45.1 | 63.2 | 77.0 | 82.5 | 47.6 | 75.2 | 86.3 | 90.2 |
| PCB-PAST (Zhang et al., 2019b) (ICCV'19) | 54.3 | 72.4 | - | - | 54.6 | 78.4 | - | - |
| SSG (Yang et al., 2019) (ICCV'19) | 53.4 | 73.0 | 80.6 | 83.2 | 58.3 | 80.0 | 90.0 | 92.4 |
| Co-teaching (Han et al., 2018)-500 (ResNet-50) | 55.7 | 71.9 | 83.5 | 88.1 | 65.1 | 82.5 | 91.8 | 93.4 |
| Co-teaching (Han et al., 2018)-500 (IBN-ResNet-50) | 61.7 | 77.6 | 88.0 | 90.7 | 71.7 | 87.8 | 95.0 | 96.5 |
| Pre-trained (ResNet-50) | 29.6 | 46.0 | 61.5 | 67.2 | 31.8 | 61.9 | 76.4 | 82.2 |
| Proposed MMT-500 (ResNet-50) | 63.1 | 76.8 | 88.0 | 92.2 | **71.2** | **87.7** | **94.9** | **96.9** |
| Proposed MMT-700 (ResNet-50) | **65.1** | **78.0** | **88.8** | **92.5** | 69.0 | 86.8 | 94.6 | **96.9** |
| Proposed MMT-900 (ResNet-50) | 63.1 | 77.4 | 88.1 | **92.5** | 66.2 | 86.8 | **94.9** | 96.6 |
| Pre-trained (IBN-ResNet-50) | 35.4 | 54.0 | 67.7 | 72.9 | 35.6 | 65.3 | 79.7 | 84.3 |
| Proposed MMT-500 (IBN-ResNet-50) | 65.7 | 79.3 | 89.1 | 92.4 | **76.5** | 90.9 | 96.4 | 97.9 |
| Proposed MMT-700 (IBN-ResNet-50) | **68.7** | **81.8** | **91.2** | **93.4** | 74.5 | 91.1 | **96.5** | **98.2** |
| Proposed MMT-900 (IBN-ResNet-50) | 67.3 | 80.8 | 90.3 | 93.0 | 72.7 | **91.2** | 96.3 | 98.0 |
| Methods | Market-to-MSMT | | | | Duke-to-MSMT | | | |
| | mAP | top-1 | top-5 | top-10 | mAP | top-1 | top-5 | top-10 |
| PTGAN (Wei et al., 2018) (CVPR'18) | 2.9 | 10.2 | - | 24.4 | 3.3 | 11.8 | - | 27.4 |
| ENC (Zhong et al., 2019) (CVPR'19) | 8.5 | 25.3 | 36.3 | 42.1 | 10.2 | 30.2 | 41.5 | 46.8 |
| SSG (Yang et al., 2019) (ICCV'19) | 13.2 | 31.6 | - | 49.6 | 13.3 | 32.2 | - | 51.2 |
| Pre-trained (ResNet-50) | 7.1 | 19.4 | 28.9 | 34.2 | 9.4 | 27.0 | 38.1 | 43.7 |
| Proposed MMT-500 (ResNet-50) | 16.6 | 37.5 | 50.6 | 56.5 | 17.9 | 41.3 | 54.2 | 59.7 |
| Proposed MMT-1000 (ResNet-50) | 21.6 | 46.1 | 59.8 | 66.1 | **23.5** | 50.0 | 63.6 | 69.2 |
| Proposed MMT-1500 (ResNet-50) | **22.9** | **49.2** | **63.1** | **68.8** | 23.3 | **50.1** | **63.9** | **69.8** |
| Proposed MMT-2000 (ResNet-50) | 20.8 | 45.7 | 59.6 | 65.6 | 22.4 | 49.0 | 62.5 | 67.8 |
| Pre-trained (IBN-ResNet-50) | 9.5 | 25.3 | 36.2 | 41.6 | 11.9 | 32.6 | 44.7 | 50.4 |
| Proposed MMT-500 (IBN-ResNet-50) | 19.6 | 43.3 | 56.1 | 61.6 | 23.3 | 50.0 | 62.8 | 68.4 |
| Proposed MMT-1000 (IBN-ResNet-50) | 26.3 | 52.5 | 66.3 | 71.7 | **29.7** | **58.8** | 71.0 | 76.1 |
| Proposed MMT-1500 (IBN-ResNet-50) | **26.6** | **54.4** | **67.6** | **72.9** | 29.3 | 58.2 | **71.6** | **76.8** |
| Proposed MMT-2000 (IBN-ResNet-50) | 25.1 | 52.7 | 65.9 | 71.3 | 28.1 | 56.8 | 70.8 | 76.0 |

Table 1: Experimental results of the proposed MMT and state-of-the-art methods on Market-1501 (Zheng et al., 2015), DukeMTMC-reID (Ristani et al., 2016), and MSMT17 (Wei et al., 2018) datasets, where MMT-$M_t$ represents the result with $M_t$ pseudo classes. Note that none of $M_t$ values equals the actual number of identities but our method still outperforms all state-of-the-arts.

sets are different from $M_t$. We test different $M_t$ values that are either smaller or greater than actual numbers.

## 4.3 COMPARISON WITH STATE-OF-THE-ARTS

We compare our proposed MMT framework with state-of-the-art methods on the four domain adaptation tasks, Market-to-Duke, Duke-to-Market, Market-to-MSMT and Duke-to-MSMT. The results are shown in Table 1. Our MMT framework significantly outperforms all existing approaches with both ResNet-50 and IBN-ResNet-50 backbones, which verifies the effectiveness of our method. Moreover, we almost approach fully-supervised learning performances (Sun et al., 2018; Ge et al., 2018) without any manual annotations on the target domain. No post-processing technique, *e.g.* re-ranking (Zhong et al., 2017a) or multi-query fusion (Zheng et al., 2015), is adopted.

Specifically, by adopting the ResNet-50 (He et al., 2016) backbone, we surpass the state-of-the-art clustering-based SSG (Yang et al., 2019) by considerable margins of 11.7% and 12.9% mAP on Market-to-Duke and Duke-to-Market tasks with simpler network architectures and lower output feature dimensions. Furthermore, evident 9.7% and 10.2% mAP gains are achieved on Market-to-MSMT and Duke-to-MSMT tasks. Recall that $M_t$ is the number of clusters or number of hard pseudo labels manually specified. More importantly, we achieve state-of-the-art performances on all tested target datasets with different $M_t$, which are either fewer or more than the actual number of identities in the training set of the target domain. Such results prove the necessity and effectiveness of our proposed pseudo label refinery for hard pseudo labels with inevitable noises.

| Duke-to-Market | IBN-ResNet-50 | | | | ResNet-50 | | | |
|---|---|---|---|---|---|---|---|---|
| | mAP | top-1 | top-5 | top-10 | mAP | top-1 | top-5 | top-10 |
| Pre-trained (only $\mathcal{L}_{id}^s$ & $\mathcal{L}_{tri}^s$) | 35.6 | 65.3 | 79.7 | 84.3 | 31.8 | 61.9 | 76.4 | 82.2 |
| Baseline (only $\mathcal{L}_{id}^t$ & $\mathcal{L}_{tri}^t$) | 62.7 | 84.4 | 92.7 | 95.5 | 53.5 | 76.0 | 88.1 | 91.9 |
| Baseline+MMT-500 (only $\mathcal{L}_{sid}^t$ & $\mathcal{L}_{stri}^t$) | 34.5 | 59.7 | 73.0 | 78.0 | 22.4 | 46.5 | 61.5 | 67.4 |
| Baseline+MMT-500 (w/o $\mathcal{L}_{id}^t$) | 38.0 | 63.4 | 74.9 | 79.4 | 24.9 | 50.3 | 64.0 | 69.8 |
| Baseline+MMT-500 (w/o $\mathcal{L}_{tri}^t$) | 76.2 | 90.8 | **96.6** | **97.9** | 72.0 | 87.8 | **95.5** | **96.9** |
| Baseline+MMT-500 (w/o $\mathcal{L}_{sid}^t$) | 69.6 | 87.4 | 95.2 | 96.7 | 62.6 | 84.0 | 93.4 | 95.4 |
| Baseline+MMT-500 (w/o $\mathcal{L}_{stri}^t$) | 71.7 | 88.5 | 95.1 | 96.6 | 65.9 | 84.0 | 93.1 | 95.5 |
| Baseline+MMT-500 (w/o $\theta_2$) | 72.8 | 89.1 | 95.2 | 97.1 | 67.5 | 86.1 | 94.3 | 96.1 |
| Baseline+MMT-500 (w/o $E[\theta]$) | 72.1 | 88.7 | 95.4 | 97.3 | 62.3 | 80.5 | 91.3 | 94.0 |
| Baseline+MMT-500 | **76.5** | **90.9** | 96.4 | 97.9 | 71.2 | 87.7 | 94.9 | **96.9** |
| Market-to-Duke | IBN-ResNet-50 | | | | ResNet-50 | | | |
| | mAP | top-1 | top-5 | top-10 | mAP | top-1 | top-5 | top-10 |
| Pre-trained (only $\mathcal{L}_{id}^s$ & $\mathcal{L}_{tri}^s$) | 35.4 | 54.0 | 67.7 | 72.9 | 29.6 | 46.0 | 61.5 | 67.2 |
| Baseline (only $\mathcal{L}_{id}^t$ & $\mathcal{L}_{tri}^t$) | 55.0 | 72.3 | 84.4 | 88.1 | 48.2 | 66.4 | 79.8 | 84.0 |
| Baseline+MMT-500 (only $\mathcal{L}_{sid}^t$ & $\mathcal{L}_{stri}^t$) | 24.5 | 38.0 | 50.1 | 56.1 | 13.6 | 24.3 | 36.4 | 42.5 |
| Baseline+MMT-500 (w/o $\mathcal{L}_{id}^t$) | 27.5 | 42.0 | 53.9 | 60.3 | 15.3 | 25.8 | 37.7 | 43.7 |
| Baseline+MMT-500 (w/o $\mathcal{L}_{tri}^t$) | 65.6 | **79.4** | **89.8** | 92.3 | 63.0 | **77.3** | **88.3** | 91.6 |
| Baseline+MMT-500 (w/o $\mathcal{L}_{sid}^t$) | 60.3 | 75.7 | 86.6 | 89.9 | 58.1 | 74.9 | 85.2 | 89.5 |
| Baseline+MMT-500 (w/o $\mathcal{L}_{stri}^t$) | 61.7 | 77.1 | 86.5 | 89.6 | 59.5 | 73.9 | 85.5 | 88.8 |
| Baseline+MMT-500 (w/o $\theta_2$) | 62.1 | 77.6 | 86.8 | 89.7 | 58.2 | 74.1 | 86.0 | 89.3 |
| Baseline+MMT-500 (w/o $E[\theta]$) | 61.1 | 76.3 | 86.6 | 89.8 | 55.7 | 70.0 | 83.6 | 87.2 |
| Baseline+MMT-500 | **65.7** | 79.3 | 89.1 | **92.4** | **63.1** | 76.8 | 88.0 | **92.2** |

Table 2: Ablation studies of our proposed MMT on Duke-to-Market and Market-to-Duke tasks with $M_t$ of 500. Note that the actual numbers of identities are not equal to 500 for both datasets but our MMT method still shows significant improvements.

To compare with relevant methods for tackling general noisy label problems, we implement Co-teaching (Han et al., 2018) on unsupervised person re-ID task with 500 pseudo identities on the target domain, where the noisy labels are generated by the same clustering algorithm as our MMT framework. The hard classification (cross-entropy) loss is adopted on selected clean batches. All the hyper-parameters are set as the same for fair comparison, and the experimental results are denoted as "Co-teaching (Han et al., 2018)-500" with both ResNet-50 and IBN-ResNet-50 backbones in Table 1. Comparing "Co-teaching (Han et al., 2018)-500 (ResNet-50)" with "Proposed MMT-500 (ResNet-50)", we observe significant 7.4% and 6.1% mAP drops on Market-to-Duke and Duke-to-Market tasks respectively, since Co-teaching (Han et al., 2018) is designed for general close-set recognition problems with manually generated label noise, which could not tackle the real-world challenges in unsupervised person re-ID. More importantly, it does not explore how to mitigate the label noise for the triplet loss as our method does.

## 4.4   Ablation Studies

In this section, we evaluate each component in our proposed framework by conducting ablation studies on Duke-to-Market and Market-to-Duke tasks with both ResNet-50 (He et al., 2016) and IBN-ResNet-50 (Pan et al., 2018) backbones. Results are shown in Table 2.

**Effectiveness of the soft pseudo label refinery.** To investigate the necessity of handling noisy pseudo labels in clustering-based UDA methods, we create baseline models that utilize only off-line refined hard pseudo labels, *i.e.*, optimizing equation 9 with $\lambda_{id}^t = \lambda_{tri}^t = 0$ for the two-step training strategy in Section 3.1. The baseline model performances are present in Table 2 as "Baseline (only $\mathcal{L}_{id}^t$ & $\mathcal{L}_{tri}^t$)". Considerable drops of 17.7% and 14.9% mAP are observed on ResNet-50 for Duke-to-Market and Market-to-Duke tasks. Similarly, 13.8% and 10.7% mAP decreases are shown on the IBN-ResNet-50 backbone. Stable increases achieved by the proposed on-line refined soft pseudo labels on different datasets and backbones demonstrate the necessity of soft pseudo label refinery and the effectiveness of our proposed MMT framework.

**Effectiveness of the soft softmax-triplet loss.** We also verify the effectiveness of soft softmax-triplet loss with softly refined triplet labels in our proposed MMT framework. Experiments of removing the soft softmax-triplet loss, *i.e.*, $\lambda_{tri}^t = 0$ in equation 9, but keeping the hard softmax-triplet loss (equation 6) are conducted, which are denoted as "Baseline+MMT-500 (w/o $\mathcal{L}_{stri}^t$)". All experiments without the supervision of soft triplet loss show distinct drops on Duke-to-Market and Market-to-Duke tasks, which indicate that the hard pseudo label with hard triplet loss hinders the feature learning capability because it ignores pseudo label noise by the clustering algorithms.

Specifically, the mAP drops are 5.3% on ResNet-50 and 4.8% on IBN-ResNet-50 when evaluating on the target dataset Market-1501. As for the Market-to-Duke task, similar mAP drops of 3.6% and 4.0% on the two network structures can be observed. An evident improvement of up to 5.3% mAP demonstrates the usefulness of our proposed soft softmax-triplet loss.

**Effectiveness of Mutual Mean-Teaching.** We propose to generate on-line refined soft pseudo labels for one network with the predictions of the past average model of the other network in our MMT framework, *i.e.*, the soft labels for network 1 are output from the average model of network 2 and vice versa. We observe that the soft labels generated in such manner are more reliable due to the better decoupling between the past temporally average models of the two networks. Such a framework could effectively avoid bias amplification even when the networks have much erroneous outputs in the early training epochs. There are two possible simplification our MMT framework with less de-coupled structures. The first one is to keep only one network in our framework and use its past temporal average model to generate soft pseudo labels for training itself. Such experiments are denoted as "Baseline+MMT-500 (w/o $\theta_2$)". The second simplification is to naïvely use one network's current-iteration predictions as the soft pseudo labels for training the other network and vice versa, *i.e.*, $\alpha = 0$ for equation 4. This set of experiments are denoted as "Baseline+MMT-500 (w/o $E[\theta]$)". Significant mAP drops compared to our proposed MMT could be observed in the two sets of experiments, especially when using the ResNet-50 backbone, *e.g.* the mAP drops by 8.9% on Duke-to-Market task when removing past average models. This validates the necessity of employing the proposed mutual mean-teaching scheme for providing more robust soft pseudo labels. In despite of the large margin of performance declines when removing either the peer network or the past average model, our proposed MMT outperforms the baseline model significantly, which further demonstrates the importance of adopting the proposed on-line refined soft pseudo labels.

**Necessity of hard pseudo labels in proposed MMT.** Despite the robust soft pseudo labels bring significant improvements, the noisy hard pseudo labels are still essential to our proposed framework, since the hard classification loss $\mathcal{L}_{id}^t$ is the foundation for capturing the target-domain data distributions. To investigate the contribution of $\mathcal{L}_{id}^t$ in the final training objective function as equation 9, we conduct two experiments. (1) "Baseline+MMT-500 (only $\mathcal{L}_{sid}^t$ & $\mathcal{L}_{stri}^t$)" by removing both hard classification loss and hard triplet loss with $\lambda_{id}^t = \lambda_{tri}^t = 1$; (2)"Baseline+MMT-500 (w/o $\mathcal{L}_{id}^t$)" by removing only hard classification loss with $\lambda_{id}^t = 1$. As illustrated in Table 2, the above two experiments both result in much lower performances than the model pre-trained on the source domain ("Pre-trained (only $\mathcal{L}_{id}^s$ & $\mathcal{L}_{tri}^s$)"), which effectively validate the necessity of $\mathcal{L}_{id}^t$. The initial network usually outputs uniform probabilities for each identity, which act as soft labels for soft classification loss, since it could not correctly distinguish between different identities on the target domain. Directly training with such smooth and noisy soft pseudo labels, the networks in our framework would soon collapse due to the large bias. One-hot hard labels for classification loss are critical for learning discriminative representations on the target domain. In contrast, the hard triplet loss $\mathcal{L}_{tri}^t$ is not absolutely necessary in our framework, as experiments without $\mathcal{L}_{tri}^t$, denoted as "Baseline+MMT-500 (w/o $\mathcal{L}_{tri}^t$)" with $\lambda_{tri}^t = 1.0$, show similar performances as our final results with $\lambda_{tri}^t = 0.8$. It is much easier to learn to predict robust soft labels for the soft softmax-triplet loss in equation 8 even at early training epochs, which has only two classes, *i.e.*, positive and negative.

## 5 Conclusion

In this work, we propose an unsupervised Mutual Mean-Teaching (MMT) framework to tackle the problem of noisy pseudo labels in clustering-based unsupervised domain adaptation methods for person re-ID. The key is to conduct pseudo label refinery to better model inter-sample relations in the target domain by optimizing with the off-line refined hard pseudo labels and on-line refined soft pseudo labels in a collaborative training manner. Moreover, a novel soft softmax-triplet loss is proposed to support learning with softly refined triplet labels for optimal performances. Our method significantly outperforms all existing person re-ID methods on domain adaptation task with up to 18.2% improvements.

### Acknowledgments

This work is supported by the General Research Fund sponsored by the Research Grants Council of Hong Kong (*Nos.* CUHK14208417, CUHK14239816, CUHK14207319), the Hong Kong Innovation and Technology Support Program (*No.* ITS/312/18FX).

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

## A  APPENDIX

### A.1  FUNCTIONS OF TEMPORAL AVERAGE MODELS IN MMT

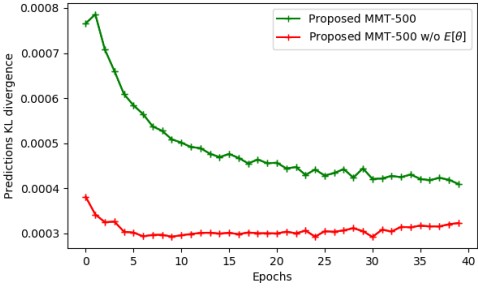

Figure 3: The predictions of temporal average models (denoted as "Proposed MMT-500") serve as more complementary and robust soft pseudo labels than those of ordinary networks (denoted as "Proposed MMT-500 (w/o $E[\boldsymbol{\theta}]$)").

Two temporal average models are introduced in our proposed MMT framework to provide more complementary soft labels and avoid training error amplification. Such average models are more

de-coupled by ensembling the past parameters and provide more independent predictions, which is ignored by previous methods with peer-teaching strategy (Han et al., 2018; Wang et al., 2019; Zhang et al., 2018b). Despite we have verified the effectiveness of such design in Table 2 by removing the temporal average model, denoted as "Baseline+MMT-500 (w/o $E[\boldsymbol{\theta}]$)", we would like to visualize the training process by plotting the KL divergence between peer networks' predictions for further comparison. As illustrated in Figure 3, the predictions by two temporal average models ("Proposed MMT-500") always keep a larger distance than predictions by two ordinary networks ("Proposed MMT-500 (w/o $E[\boldsymbol{\theta}]$)"), which indicates that the temporal average models could prevent the two networks in our MMT from converging to each other soon under the collaborative training strategy.

## A.2 PARAMETER ANALYSIS

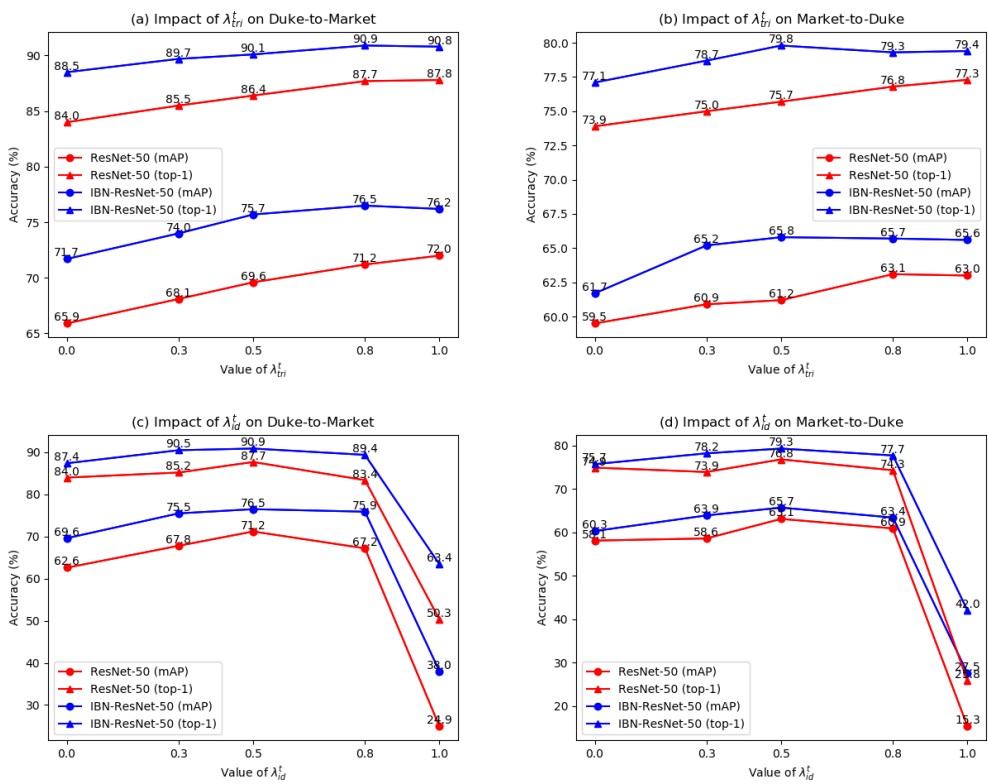

Figure 4: Performance evaluation of our proposed MMT-500 with different values of $\lambda_{tri}^t$ and $\lambda_{id}^t$ in equation 9 on Duke-to-Market and Market-to-Duke tasks in terms of mAP(%) and top-1(%) accuracies. Weighting factors $\lambda_{tri}^t$ and $\lambda_{id}^t$ balance the contributions between hard and soft pseudo labels. Specifically, only hard labels are adopted when the weighting factors are set to $0.0$, and only soft labels are utilized when the weighting factors are set to $1.0$.

We utilize weighting factors of $\lambda_{tri}^t = 0.8$, $\lambda_{id}^t = 0.5$ in all our experiments by tuning on Duke-to-Market task with IBN-ResNet-50 backbone and 500 pseudo identities. To further analyse the impact of different $\lambda_{tri}^t$ and $\lambda_{id}^t$ on different tasks, we conduct comparison experiments by varying the value of one parameter and keep the others fixed. Our MMT framework is robust and insensitive to different parameters except when the hard classification loss is eliminated with $\lambda_{id}^t = 1.0$.

**The weighting factor of hard and soft triplet losses $\lambda_{tri}^t$.** In Figure 4 (a-b), we investigate the effect of the weighting factor $\lambda_{tri}^t$ in equation 9, where the weight for soft softmax-triplet loss is $\lambda_{tri}^t$ and the weight for hard triplet loss is $(1 - \lambda_{tri}^t)$. We test our proposed MMT-500 with both ResNet-50 and IBN-ResNet-50 backbones when $\lambda_{tri}^t$ is varying from 0.0, 0.3, 0.5, 0.8 and 1.0. Specifically, the soft softmax-triplet loss is removed from the final training objective (equation 9) when $\lambda_{tri}^t$ is equal to 0.0, and the hard triplet loss is eliminated when $\lambda_{tri}^t$ is set to 1.0. We observe

that the accuracies are almost in direct ratio to the value of $\lambda^t_{tri}$ which indicate the effectiveness of our proposed novel soft softmax-triplet loss. MMT-500 achieves optimal performances with ResNet-50 backbone on both two tasks when $\lambda^t_{tri} = 1.0$. With the backbone of IBN-ResNet-50, MMT-500 obtains the best results with $\lambda^t_{tri} = 0.8$ on Duke-to-Market and $\lambda^t_{tri} = 0.5$ on Market-to-Duke. Despite the performances vary with different values of $\lambda^t_{tri}$, all the results by our method outperform state-of-the-arts significantly.

**The weighting factor of hard and soft classification losses** $\lambda^t_{id}$. Similar to the comparisons of $\lambda^t_{tri}$, we evaluate our proposed MMT-500 framework with different values of $\lambda^t_{id}$, which is the weighting factor for hard and soft classification losses in equation 9. As illustrated in Figure 4 (c-d), we observe considerable declines when the hard classification loss equation 1 is eliminated with $\lambda^t_{id} = 1.0$. Hard classification loss is essential to our proposed framework, which is fully analysed in Section 4.4. We achieve the optimal performances on both two tasks when $\lambda^t_{id} = 0.5$, while all the experiments with $\lambda^t_{id} < 1$ outperform state-of-the-arts by large margins.

