# OpenReview forum: "Mutual Mean-Teaching: Pseudo Label Refinery for Unsupervised Domain Adaptation on Person Re-identification"
_ICLR.cc/2020/Conference — Accept (Poster)_

### Official Review · AnonReviewer2 · 2019-10-13
**Official Blind Review #2**

**Rating:** 6

**Review:**

The authors' response addressed my concerns. After reading the reviews and the comments, I choose to stand with the other reviewers.

===================

This paper uses mean-teacher to ease the noisy pseudo label of clustering methods for domain adaptive Person re-identification task. The authors also propose a variant of triplet loss for soft labels. Experiments show they achieve considerable improvement over state-of-the-art methods.

Questions:
1. What's the difference between net 1 and net 2 in Fig. (2)? It seems they are redundant.

2. The results in Table (1) seems to indicate that, in MSMT, if M_t is set to be near the actual identity numbers (1041), the performance will be much better. This makes me suspect that the proposed method benefits from ground truth information of the target domain, which makes the comparison unfair.


**Experience Assessment:**

I do not know much about this area.

**Review Assessment: Checking Correctness Of Derivations And Theory:**

N/A

**Review Assessment: Checking Correctness Of Experiments:**

I assessed the sensibility of the experiments.

**Review Assessment: Thoroughness In Paper Reading:**

I made a quick assessment of this paper.

---

> ### Author Response · Authors · 2019-11-15
> **Response to Review #2**
>
> Thank you for the constructive comments! We really appreciate the comments for improving the clarity of statements and experimental verifications. The manuscript is revised accordingly, and the main concerns are listed as below.
>
>
> [QUESTION 1]
> What's the difference between Net 1 and Net 2 in Figure 2?
>
> [RESPONSE 1]
> The main differences lie in three aspects.
> (1) The two networks are initialized with different weights.
> (2) Image batches are fed into the two networks with separately random perturbations,
> (3) The two networks are optimized under different supervisions, i.e. regularized by separate soft pseudo labels which are generated by the other network.
>
>
> [QUESTION 2]
> It seems Net 1 and Net 2 are redundant.
>
> [RESPONSE 2]
> No, they are not. The two networks are complementary due to the differences introduced in [RESPONSE 1]. With the proposed mutual mean-teaching scheme, the two networks can continuously learn different knowledge from the soft pseudo labels generated by the other networks. Such design effectively refines the pseudo label noise in the unsupervised person re-ID task.
>
> Experimental results also verify the effectiveness of learning from dual networks. We conduct sufficient ablation studies in Table 2. Specifically, “Baseline+MMT-500(w/o $\theta_2$)” denotes the experiment with only one network left which is supervised by its own temporal average model. Significant performance drops from 63.1% to 58.2% in terms of mAP on Market-to-Duke task with ResNet-50 are observed by keeping only one network in our proposed framework.
>
> The effectiveness of collaborative training with multiple networks have also been verified in more general tasks.
> (1) DualNet (Hou et al., 2017) improved the image recognition accuracies by learning complementary and richer features with double feature encoders.
> (2) Deep mutual learning (Zhang et al., 2018b) performed model distillation between a pool of simple student networks and achieved better performances than the typical one-way transferring from a pre-trained static large (teacher) network.
> (3) For semi-supervised learning, Dual Student (Ke et al., 2019) broke the performance bottleneck of conventional teacher-student network by introducing to adopt two student networks with more de-coupled parameters for peer teaching.
> (4) Similar pipelines are also employed on noisy labels, for instance, Co-teaching (Han et al., 2018) simultaneously trained two same networks and conducted noisy labels detection by selecting clean data for each other.
>
> However, existing methods with two peer networks mostly focused on close-set recognition problems and could not tackle the challenges in the task of open-set unsupervised person re-ID studied in our paper. More importantly, compared to the partial or noisy labels provided by their tasks, the unsupervised person re-ID remains a more challenging problem with no label at all.
>
> References:
> 1. Saihui Hou, Xu Liu, and Zilei Wang. Dualnet: Learn complementary features for image recognition. In ICCV, 2017.
> 2. Ying Zhang, Tao Xiang, Timothy M Hospedales, and Huchuan Lu. Deep mutual learning. In CVPR, 2018b.
> 3. Zhanghan Ke, Daoye Wang, Qiong Yan, Jimmy Ren, and Rynson WH Lau. Dual student: Breaking the limits of the teacher in semi-supervised learning. In ICCV, 2019.
> 4. Bo Han, Quanming Yao, Xingrui Yu, Gang Niu, Miao Xu, Weihua Hu, Ivor Tsang, and Masashi Sugiyama. Co-teaching: Robust training of deep neural networks with extremely noisy labels. In NIPS, 2018.
>
>
> [QUESTION 3]
> The results of MSMT17 in Table 1 indicates the performance is better when $M_t$ is near the actual identity number. The proposed method benefits from ground-truth information of the target domain, which makes the comparison unfair.
>
> [RESPONSE 3]
> We conducted additional experiments with different values of $M_t$ on MSMT17 dataset during the rebuttal period. Results of the newly added Market-to-MSMT17 task in Table 1 show that the best performances are achieved when $M_t = 1500$, which is much larger than the actual identity numbers (1041). Besides, we outperform state-of-the-arts significantly in all experiments regardless of which $M_t$ value is used.
>
> We didn't intentionally set $M_t$ to be close to the actual identity number. Results are robust when $M_t$ changes within a large range.
>
> Market-to-MSMT (mAP, top-1, top-5, top-10):
> MMT-1000 (ResNet-50): 21.6, 46.1, 59.8, 66.1
> MMT-1500 (ResNet-50): 22.9, 49.2, 63.1, 68.8
> MMT-2000 (ResNet-50): 20.8, 45.7, 59.6, 65.6
>
> Duke-to-MSMT (mAP, top-1, top-5, top-10):
> MMT-1000 (ResNet-50): 23.5, 50.0, 63.6, 69.2
> MMT-1500 (ResNet-50): 23.3, 50.1, 63.9, 69.8
> MMT-2000 (ResNet-50): 22.4, 49.0, 62.5, 67.8

---

### Official Review · AnonReviewer1 · 2019-10-24
**Official Blind Review #1**

**Rating:** 8

**Review:**

After reading the reviews and the comments, I confirm my rating.

=================

The paper proposes an unsupervised framework to address the problem of noisy pseudo labels in clustering-based unsupervised domain adaptation (UDA) for person re-identification. The noise derives from the limited transferability of source-domain features, the unknown number of target-domain identities, and the imperfect results of the clustering algorithm.

The proposed framework, Mutual Mean-Teaching (MMT), performs pseudo label reﬁnery by optimizing the neural networks under the joint supervisions of off-line reﬁned hard pseudo labels and on-line reﬁned soft pseudo labels. Inspired by the teacher-student approaches (Reference: Tarvainen & Valpola, 2017; Reference: Zhang et al., 2018b), the proposed MMT framework provides robust soft pseudo labels in an on-line peer-teaching manner to simultaneously train two same networks. The networks gradually capture target-domain data distributions and thus reﬁne pseudo labels for better feature learning.

The main contribution is proposing an unsupervised framework (MMT) capable of tackling the noise problem in state-of-art UDA methods for person re-identification, via producing reliable soft labels in order to achieve better performance. Since the conventional triplet loss cannot properly work with soft labels, a softmax-triplet loss is proposed to enable training with soft triplet labels for mitigating the pseudo label noise.

The proposed MMT is evaluated on Market1501, DukeMTMC-reID, and MSMT17 datasets with four adaptation tasks: Market-to-Duke, Duke-to-Market, Market-to-MSMT, and Duke-to-MSMT. It outperforms the state-of-the-art methods with significant improvements in terms of Mean average precision (mAP) and Cumulative matching characteristic (CMC). In addition to that, ablation studies conducted to evaluate each component in the proposed MMT framework.

**Experience Assessment:**

I have read many papers in this area.

**Review Assessment: Checking Correctness Of Derivations And Theory:**

I assessed the sensibility of the derivations and theory.

**Review Assessment: Checking Correctness Of Experiments:**

I assessed the sensibility of the experiments.

**Review Assessment: Thoroughness In Paper Reading:**

I read the paper thoroughly.

---

> ### Author Response · Authors · 2019-11-15
> **Response to Review #1**
>
> Thank you for the positive and helpful comments!

---

### Official Review · AnonReviewer3 · 2019-10-24
**Official Blind Review #3**

**Rating:** 6

**Review:**

This paper proposes an unsupervised domain adaptation method for person re-identification. The proposed method handles noises on pseudo labels created by unsupervised clustering. Two networks are used for training, and the estimated confidences of other models are used for the next training iterations. The temporally average model is used for each network to avoid error amplification. Also, soft softmax-triplet loss is proposed to handle soft labels for triplet loss.

The handling label noises in unsupervised domain adaptation on person re-identification are new. The proposed model produces very high performance and the contribution for person re-identification community is good.

However, I would like to see more insights into the proposed model for the contribution of the general deep learning conference.

First, this paper lacks a survey of works on handling label noises. For example,
B.Han, Q.Yao, X.Yu, G.Niu, M.Xu, W.Hu, I.Tsang, M.Sugiyama, Co-teaching: Robust Training of Deep Neural Networks with Extremely Noisy Labels, NeurIPS2018.

I could not fully understand why the temporal averaging of model parameters prevents the two models from being the same. I would like to see a theoretical explanation or experimental evidence for this claim.

The proposed method also uses noisy hard pseudo labels for training, as shown in Eq.(9).
Why are the noisy hard labels used? What is the performance when only soft labels are used for model updates?

In the experiment, \lambda^t_{id} = 0.5, \lambda^t_{tri} = 0.8 are used. Why these parameters are different between softmax and triplet losses?

p.1 (Zhang et al., 2018b) and p.5 (Zhang et al., 2019a) are missing in references.




**Experience Assessment:**

I have read many papers in this area.

**Review Assessment: Checking Correctness Of Derivations And Theory:**

I assessed the sensibility of the derivations and theory.

**Review Assessment: Checking Correctness Of Experiments:**

I assessed the sensibility of the experiments.

**Review Assessment: Thoroughness In Paper Reading:**

I read the paper at least twice and used my best judgement in assessing the paper.

---

> ### Author Response · Authors · 2019-11-15
> **Response to Review #3**
>
> Thank you for the constructive comments! We really appreciate the comments for improving the clarity of statements and experimental verifications. The manuscript is revised accordingly, and the responses to your main concerns are listed below.
>
>
> [QUESTION 1]
> The paper lacks a survey of works on handling noisy labels.
>
> [RESPONSE 1]
> We incorporate more related works on handling noisy labels in Section 2 marked as blue. Besides, we also experimentally compared with the most relevant method, Co-teaching (Han et al., NIPS 2018), mentioned by the reviewer on unsupervised person re-ID settings. The results in Table 1 show that our method is significantly better than Co-teaching by at least 6% mAP with the same ResNet-50 backbone.
>
> Implementation details and further analysis are presented in Section 4.3. Co-teaching could not tackle the real-world challenges in unsupervised person re-ID since it is designed for general close-set recognition problems with manually generated label noise. More importantly, it does not explore how to mitigate the label noises for the triplet loss as our method does.
>
>
> [QUESTION 2]
> Why the temporal averaging of model parameters prevents the two models from being the same?
>
> [RESPONSE 2]
> Intuitively, when removing temporal average models, Net 1 is directly trained to approach the predictions of Net 2 with on-the-fly parameters $\theta_2^{(T-1)}$ at iteration $(T-1)$.
> In our MMT, Net 1 is optimized toward the predictions of Net 2 with averaging model parameters $E^{(T)}[\theta_2]$ which is ensembled from iteration $(0)$ to iteration $(T-1)$.
> Obviously, temporal average models provide more complementary and independent predictions which act as more robust soft pseudo labels in our framework, since the parameters are more de-coupled with larger time intervals.
>
> Experimental evidences are shown in Section A.1. The KL-divergence between predictions of two temporal averaging models is much larger than that of two plain networks during the training.
>
>
> [QUESTION 3]
> Why are the noisy hard labels used? What is the performance when only soft labels are used for model updates?
>
> [RESPONSE 3]
> The noisy hard pseudo labels are utilized since the hard classification loss is the foundation for capturing the target-domain data distributions. The soft labels for classification loss are almost uniform and uninformative at early epochs, since the initial network could not correctly distinguish between different identities on the target domain. Independently training with such smooth and noisy soft pseudo labels, the networks in our framework would soon collapse due to the large bias.
>
> With regularization by only soft pseudo labels, i.e. removing both the hard classification loss and the hard triplet loss, the framework totally fails with performances even lower than the pre-trained model on the source domain (new result in Table 2 denoted as “Baseline+MMT-500 (only $\mathcal{L}^t_{sid}$ & $\mathcal{L}^t_{stri}$)”).
>
> We further investigate the contributions of the hard classification loss and hard triplet loss in our proposed framework separately. The experimental results are shown in Table 2 (marked in blue) and we observe that the hard classification loss is essential to our framework while the hard triplet loss is not absolutely necessary. Detailed analysis and explanations are listed in Section 4.4, namely “Necessity of hard pseudo labels in proposed MMT”.
>
>
> [QUESTION 4]
> Why the weighting factors are different between softmax and triplet losses?
>
> [RESPONSE 4]
> We empirically searched the weights $\lambda^t_{tri}$ and $\lambda^t_{id}$ on the Duke-to-Market task with $M_t=500$ pseudo classes and IBN-ResNet-50 backbone. The searched hyper-parameters are directly used in other settings.
>
> From our observation, we found that the soft softmax-triplet loss has larger weight than the soft classification loss, since it is much easier to predict robust soft labels for softmax-triplets, which only has two classes, i.e. positive and negative.
>
> To further investigate the influence of $\lambda^t_{tri}$ and $\lambda^t_{id}$, additional experiments are conducted in Section A.2 during rebuttal. We claim that our framework is not sensitive to the weighting factors as we outperform state-of-the-arts with all tested hyper-parameters, except when the hard classification loss is eliminated ($\lambda^t_{id}=1.0$). We have analysed the necessity of hard classification loss in [RESPONSE 3].
>
>
> [QUESTION 5]
> p.1 (Zhang et al., 2018b) and p.5 (Zhang et al., 2019a) are missing in references.
>
> [RESPONSE 5]
> These two works have been listed in the references in our initial submission.
> p.1 (Zhang et al., 2018b): “Ying Zhang, Tao Xiang, Timothy M Hospedales, and Huchuan Lu. Deep mutual learning. In CVPR, 2018b.”
> p.5 (Zhang et al., 2019a): “Ji Zhang, Yannis Kalantidis, Marcus Rohrbach, Manohar Paluri, Ahmed Elgammal, and Mohamed Elhoseiny. Large-scale visual relationship understanding. In AAAI, 2019a.”

---

### Decision · Program_Chairs · 2019-12-19

**Decision:**

Accept (Poster)

**Comment:**

The paper proposes an unsupervised framework for domain adaptation in the context of person re-identification to reduce the effect of noisy labels. They use refined soft labels and propose a soft softmax-triplet loss to support learning with these soft labels.

All reviewers have unanimously agreed to accept the paper and appreciated the comprehensive experiments on four datasets and ablation studies which give some insights about the proposed method. I agree with the assessment of the reviewers and recommend that this paper be accepted.